# Assessment of Melting Kinetics of Sugar-Reduced Silver Ear Mushroom Ice Cream under Various Additive Models

**Shu-Yao Tsai [1,*]**, **Gregory J. Tsay [2,3]**, **Chien-Yu Li [4]**, **Yu-Tzu Hung [1]** and **Chun-Ping Lin [1,5,6,*]**

1   Department of Food Nutrition and Health Biotechnology, Asia University, 500, Lioufeng Rd., Wufeng, Taichung 41354, Taiwan; hytheyla@gmail.com
2   Division of Immunology and Rheumatology, Department of Internal Medicine, China Medical University Hospital, Taichung 40447, Taiwan; jjtsay@mail.cmu.edu.tw
3   Department of Internal Medicine, School of Medicine, China Medical University Hospital, China Medical University, Taichung 40447, Taiwan
4   Department of Neurosurgery, Asia University Hospital, Taichung 41354, Taiwan; sbrain.lee@mail.cmu.edu.tw
5   Office of Environmental Safety and Health, Asia University, 500, Lioufeng Rd., Wufeng, Taichung 41354, Taiwan
6   Department of Medical Research, China Medical University Hospital, China Medical University, 91, Hsueh-Shih Rd., Taichung 40402, Taiwan
*   Correspondence: sytsai@asia.edu.tw (S.-Y.T.); chunping927@gmail.com (C.-P.L.)

**Abstract:** This study focuses on assessing the effects of various food processing silver ear (*Tremella fuciformis*) powders in sugar-reduced ice cream through melting kinetic simulation, sensory properties and functional ingredients. *T. fuciformis*, a natural anti-melting stabilizer in ice cream, has the advantage of functional ingredients. Using 100, 200, and 300 mesh of particle sizes, and then selecting a suitable particle size, those are added to the additive ratios of 0.4, 0.9, and 1.4% *T. fuciformis* powder to replace fresh *T. fuciformis* fruit body. Decreased particle size of *T. fuciformis* powder significantly increased ice cream stability. Comparisons of sensory evaluation and melting properties, in order to learn the differences of *T. fuciformis* ice cream under various stabilizer models, were evaluated and elucidated. Therefore, we obtained 300 mesh at 0.9% additive ratio of *T. fuciformis* powder, which is closest to the fresh fruit body/base ice cream. The enrichment of ice cream with *T. fuciformis* is to enhance the nutritional aspects and develop a functional food. Overall, the kinetic parameters of *T. fuciformis* ice cream melting can be provided as a reference for frozen dessert processing technology.

**Keywords:** silver ear (*Tremella fuciformis*); anti-melting stabilizer; ice cream; functional ingredient; frozen dessert processing technology

---

## 1. Introduction

Ice cream, the best-known dairy dessert, represents a complex frozen system containing air bubbles, fat globules, and ice crystals dispersed in a freeze-concentrated dispersion [1,2]. With the wide variety of ice creams, the formulation is highly dependent on the mixing ingredients. Ice cream ingredients contain milk powder, emulsifiers, stabilizers, sweeteners, and flavoring agents, while fat and sugar are the main compounds that provide the caloric content. Therefore, the aim was to reduce or replace these important ingredients in ice cream formulations in order to develop healthy products that meet consumer requirements. These ice cream products include fiber-enriched, prebiotic, low fat,

and low sugar foods [3–8]. This is because the formulation affects ice cream quality characteristics, such as sensory properties, freezing behavior, and melting properties [9–11]. The addition of stabilizers primarily improves the smooth texture, decreases the melting rate of the mixture, and promotes the formation of large ice and lactose crystals [12]. In addition to these traditional emulsifiers, numerous scientists have striven to actively develop new stabilizers like gum for ice cream [13].

Heterobasidiae edible and medicinal mushrooms are a subclass of Basidiomycetes. Among these, *Tremella fuciformis* is favorite artificially cultivated mushroom in Taiwan, known as silver ear or white jelly fungus [14]. *T. fuciformis* is a popular food and herbal medicine ingredient due to having low energy and lipid content, but it serves as a rich source of polysaccharides and dietary fiber; it is widely used for nutritive and tonic actions in Asian countries [15]. *T. fuciformis* has been used for medicinal purposes due to its possible bioactivity, which includes antioxidant properties [16], anti-inflammatory functions [17], antineoplastic [18] and immunomodulatory effects [19], and prevention of neurodegenerative disorders [20]. Fresh *T. fuciformis* supplies rapidly lose their general quality due to fruit body autolysis and high respiration rate. Therefore, drying is a comparatively effective method for the storage of mushrooms, and dehydrated mushrooms have been especially appreciated by consumers [21]. As a food additive, *T. fuciformis* powders can supplement bread [22] or can be used as meat substitutes [23] to improve their sensory and physicochemical properties. Tsai et al. [14] found that small particle sizes of *T. fuciformis* powder had water holding capacity and solubility that are well suited for the manufacture of instant foods.

We developed a special technical method melting kinetics simulation to learn the differences between various natural stabilizer models of *T. fuciformis* to add in ice cream [24]. It is a novel idea from the perspective of melting kinetics to learn the various additives that have minor physical change differences in terms of melting characteristics by melting kinetic model simulation. So far, scientists have agreed that melting or thawing is a complex physical change, but it is an interesting topic to use simple and feasible methods to evaluate melting without involving chemical changes. Anyway, the scientific theory needs to be actually combined. If you find an exceptionally suitable anti-melting additive, it has a bad taste and is in vain. To convey the principle of natural deliciousness, the ice cream must be savory, the advantage of slowing down the melting and increasing the rich taste at the same time in this study.

Therefore, we used fresh *T. fuciformis* fruit bodies, substrates, various particle sizes of dry powders, control of different additive ratios, then conducted ice cream sensory evaluation and melting rate estimation to compare with the micro thermal analysis technology. The effects of various natural stabilizers were evaluated on mouthfeel and melting reaction for *T. fuciformis* ice cream. According to melting reaction kinetics, which are comprehended in the food processing industry to apply a cross-domain approach for assessment of ice cream stabilizer effect. Previous studies by the authors used kinetic simulation methods to ascertain the differences in moisture desorption and food drying methods of mushrooms [14], but no attempt was made to explore the melting of ice cream. Worldwide, scholars have also rarely engaged in similar research. This is a novel application research on food processing innovation methods.

## 2. Materials and Methods

### 2.1. Ice Cream Mixture Formulation and Processing

Fresh *Tremella fuciformis* (TF) was donated by Mid-Sum Biotechnology Co. Ltd. (Taichung, Taiwan, ROC). At harvest, fresh mushrooms were removed from the jar and cut into fruit body (100–120 g fresh weight/mushroom) and the base was sliced (10 g fresh mass/slice). The sliced bases were freeze-dried and ground into a powder with a milling machine (Retsch Ultra Centrifugal Mill and sieving machine, Haan, Germany), screened through 100, 200, and 300 mesh sieves, and then stored at 4 °C until use. Milk cream (Taiwan Branch of Singapore Business Pan-Asia Dragon PTE. Ltd., Taipei, Taiwan), emulsifier (imported from First Chemical Co., Ltd., Taipei, Taiwan, manufactured by Shiohama Co.,

Japan), sucrose (Taiwan Sugar Co. Ltd., Tainan, Taiwan), sodium chloride (Taiyen Biotech Co., Ltd., Tainan, Taiwan), sodium carboxymethyl cellulose (imported by Da Kwan Trading Co. Ltd., New Taipei, Taiwan, manufactured by Nippon Paper Industries Co., Ltd., Tokyo, Japan), and milk powder and maltodextrin (Sally Biotech Co. Ltd., Changhua, Taiwan) were purchased from a local market.

Fresh *T. fuciformis* fruit body, *T. fuciformis* sliced base, and 100, 200, and 300 mesh *T. fuciformis* powders were used to make various ice creams, assigned as fresh fruit body (TF), fresh base (TFB), 100, 200, and 300 mesh *Tremella fuciformis* powder (TFP) (Figure 1A). The ice cream production method is summarized in Figure 1B. Experimental ice creams were formulated as reported in Table 1. The 300 mesh *T. fuciformis* powder was used to create different additive ratios in ice cream, assigned as 300 mesh TFP low ratio (0.4%), 300 mesh TFP medium ratio (0.9%), and 300 mesh TFP high ratio (1.4%). Ice cream mixes were prepared by using a batch ice cream maker (2 L capacity, Kaiser Kice-2030, China) for 60 min. Briefly, the TP sample was blended (90 °C hot water, 4 min), and then milk powder, sucrose, sodium chloride, emulsifier, and stabilizer were added and the mix was agitated for 1 min. Afterward, liquefied milk cream was added and the entire mixture was blended for 1 min using a mixer. The mixture was heated using a batch heater (30 secs at 90 °C). Then, the ice creams were packed in 500 mL paper cups and stored (−18 °C, 24 h) for hardening, followed by storing in a freezer (−18 °C) prior to analysis.

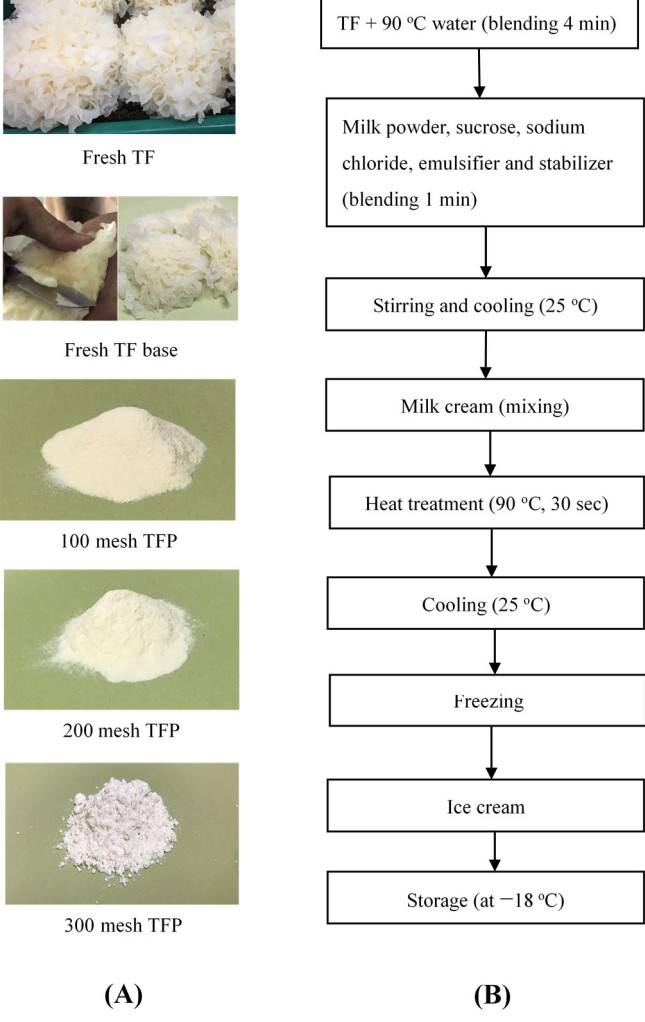

**(A)**　　　　　　　　　　　**(B)**

**Figure 1.** *Tremella fuciformis* sample (**A**) and ice cream production flow chart (**B**).

**Table 1.** *Tremella fuciformis* ice cream formulation.

| Ingredients/g | Control | Fresh TF | Fresh TF Base | 100 Mesh TFP | 200 Mesh TFP | 300 Mesh TFP-0.4% | 300 Mesh TFP-0.9% | 300 Mesh TFP-1.4% |
|---|---|---|---|---|---|---|---|---|
| Milk cream | 180 | 180 | 180 | 180 | 180 | 180 | 180 | 180 |
| Milk powder | 9 | 9 | 9 | 9 | 9 | 9 | 9 | 9 |
| Sucrose | 40 | 50 | 50 | 40 | 40 | 40 | 40 | 40 |
| Sodium chloride | 0.5 | 0.5 | 0.5 | 0.5 | 0.5 | 0.5 | 0.5 | 0.5 |
| Emulsifier | 2 | 2 | 2 | 2 | 2 | 2 | 2 | 2 |
| Sodium carboxymethyl cellulose | 0.45 | 0.45 | 0.45 | 0.45 | 0.45 | 0.45 | 0.45 | 0.45 |
| Maltodextrin | 0.45 | 0.45 | 0.45 | 0.45 | 0.45 | 0.45 | 0.45 | 0.45 |
| Water | 300 | 300 | 300 | 300 | 300 | 300 | 300 | 300 |
| Silver ear | - | 54 | 54 | 4.83 | 4.83 | 2.13 | 4.85 | 7.6 |
| Total | 532.4 | 596.4 | 596.4 | 537.25 | 537.25 | 534.55 | 537.25 | 540.0 |

## 2.2. Consumer Sensory Evaluation

The ice cream samples stored at −18 °C were tempered for 10 min at room temperature for sensory evaluation at the Asia University campus, Taichung, Taiwan. A total of 50 untrained consumers from ages 19 to 50 completed a questionnaire. Five sensory attributes (flavor, mouthfeel, texture, sweetness, and overall acceptability) were measured for the ice cream samples. The values of 1, 3, and 5 on the five-point hedonic scale represented "extremely dislike", "neither like nor dislike", and "extremely like", respectively [22].

## 2.3. Melting Behavior and Melting Rate Measurements

The behavior of ice cream to retain its shape during melting was evaluated through an image analysis method. Roughly 100 g of cylindrical blocks of conditioned ice creams (−18 °C, 24 h) was placed on an iron plate at 24 ± 0.5 °C. Photographs of the samples during the melting test were taken with a phone camera (iPhone 7 Plus, Apple Inc., USA) in standard conditions at fixed time intervals of 5 min, for a total time of 90 min. The melting characteristics, expressed as melting rate (g/min), were evaluated as described by Balthazar et al. [4]. Briefly, 50 g of ice cream (−18 °C) was placed on a strainer and left to melt into a 200 mL beaker at room temperature (23 ± 2 °C) until 100% of the sample was melted. The mass of melted ice cream was recorded every 5 min with the aim of obtaining a sigmoidal curve representing the kinetics of the melting process. From the linear region of the curve, the most likely straight line was calculated, with its slope representing the melting rate (g/min).

## 2.4. DSC Measurements

Measurements of the melting properties of *T. fuciformis* ice cream were performed by differential scanning calorimetry (DSC) (TA Q20) with a refrigerated cooling system (RCS 90) (TA Instruments, Newcastle, DE, USA). DSC analysis was performed on samples sealed in 20 μL aluminum pans; the test cell was sealed manually by a special tool equipped with the TA DSC. In all DSC studies, nitrogen was the purge gas with a flow rate of 50 mL/min. ASTM E698 was used to obtain thermal curves for analyzing the parameters. Aliquots (4.0 mg) of each sample were used for acquiring the experimental data. Non-isothermal tests of the heating rate selected for the programmed temperature ramp included 2, 4, and 6 °C/min for the range of temperature increase chosen from holding for 5 min at −60 °C, followed by heating from −60 to 30 °C for each melting experiment [25]. The melting property analysis also compared commercial ice cream and 10%, 30%, and 50% sucrose solutions in order to understand the glass transition temperature ($T_g$) characteristics of sucrose in ice cream and its melting effects with respect to ice cream, which is also to preliminarily exclude the interference of sucrose on the melting of ice cream. The DSC dynamic tests of the heating rate led to the selection of 6 °C/min for the

conditions of holding for 5 min at −60 °C and then heating from −60 to 30 °C for each glass transition temperature experiment.

## 2.5. Melting Kinetic Simulation

Comparisons of Avrami Erofeev's kinetic and the proto-kinetic equations were applied to evaluate the ice cream melting properties for the melting reaction, as follows [26]:

$$r_i = ke^{-\frac{E_a}{RT}}(1-\alpha)[-\ln(1-\alpha)]^{n_0} \text{ Avrami Erofeev's kinetic equation} \tag{1}$$

$$r_i = ke^{-\frac{E_a}{RT}}\alpha^{n_1}[1-\alpha]^{n_2} \text{ Proto-kinetic equation} \tag{2}$$

where $r_i$ is the reaction rate, $E_a$ is the apparent activation energy of melting, $k$ is the pre-exponential factor of the reaction, R is the ideal gas constant, $\alpha$ is the degree of conversion of a reaction or stage, and $n_i$ is the reaction order of melting (i = 0, 1, and 2). We developed a melting reaction for *T. fuciformis* ice cream that includes the melting characteristics, such as the reaction order ($n_i$), enthalpy of the melting reaction ($\Delta H$), and the apparent activation energy of melting ($E_a$), which could be applied in the design of processing conditions for ice cream.

## 2.6. Determination of Physical Characteristic

Physical characteristic analyses were performed on hardened ice cream sample on day 1. The pH values of the ice cream sample were measured using an S20 pH Meter (Mettler Toledo International, Inc. Greifensee, Switzerland). The ice cream samples were subjected to color measurement using an S80 color measuring system (BYK Additives & Instruments, Columbia, MD, USA). The color of the samples was expressed as *L*-value (lightness), *a*-value (redness/greenness) and *b*-value (yellowness/blueness). The whiteness index (*WI*) was calculated based on the following equation [22]:

$$WI = 100 - \sqrt{(100-L)^2 + a^2 + b^2} \tag{3}$$

The hardness values of ice cream were determined by texture profile analysis (EZ Test/CE, Shimadzu Model, Japan). For each sample, three measurements with three replicates were performed using a cylindrical probe (5 mm diameter). The hardness was expressed as the peak pressure force ($N/mm^2$) during penetration. The apparent viscosity of the sample was determined using a sine wave Vibro-viscometer A&D SV-10 (A&D Company, Limited, Tokyo, Japan), that was at constant frequency 30 Hz and amplitude less than 1 mm [14].

## 2.7. Determination of Chemical Characteristics

The ice cream samples were freeze-dried and ground into powder by using a milling machine (Retsch ultracentrifugal mill and sieving machine, Haan, Germany) and stored at 4 °C until used. Dry matter, ash, fat and protein were determined in ice cream samples according the AOAC (2000) methods. The moisture content of the ice cream samples was measured by an MB45 moisture analyzer (Ohaus Corporation, NJ, USA) at 105 °C. The ash content was analyzed by using gravimetric methods at 550 °C. The fat content of the ice cream was determined using the Soxhlet method. The protein content was assayed according to Kjeldahl method. Water-insoluble and water-soluble dietary fibers were analyzed following the AOAC 991.43 enzymatic-gravimetric method using a dietary fibers assay kit (Megazyme International Ireland Ltd., Wicklow, Ireland). The polysaccharide content was determined by phenol–sulfuric acid [14]. Ice cream sample was extracted with water in a 1: 30 (*w/w*) ratio at 121 °C and 15 min. The mixture was cooled to room temperature and filtered through Whatman No. 4 filter paper. The combined filtrate was dialyzed by using a Cellu Sep T2 tubular membrane (MWCO: 6000–8000, Membrane Filtration Products, Inc., Seguin, TX, USA) for 24 h, resulting in a water-soluble polysaccharide sample.

### 2.8. Statistical Analyses

The experimental data obtained were expressed as mean ± standard error and subjected to an analysis of variance for a completely random design using an SAS 9.4 Software (SAS Institute, Inc., Cary, NC, USA). The difference among means was determined using Duncan's multiple range tests at the level of α = 0.05.

## 3. Results and Discussion

### 3.1. Sensory Quality Characteristics of T. fuciformis Ice Cream

The taste should remain unchanged, with the adjustments being the different additives of *T. fuciformis*, such as fresh fruit body, substrate, various particle sizes, and addition ratio. Then sensory evaluation and melting rate analyses were conducted. The results of sensory quality characteristics are shown in Figure 2. The flavor, mouthfeel, and texture scores were determined in the range of 3.41–4.01 points of all ice cream samples. The highest score of all samples was found to be 4.05 in the TF base. The fresh TF base exhibited the best taste for all ice creams, next was the Fresh TF ice cream, and these were followed by the 300 mesh TFP that was added to a proportion of 0.9% in the ice cream. Meanwhile, we also compared the 100, 200, and 300 mesh *T. fuciformis* powders as natural stabilizers to be added into ice cream. Each additive amount was 0.9%, and the 300 mesh powder ice cream was determined to taste the best. The sensory evaluation of additive ratio was carried out with 300 mesh powder in the ratios of 0.4, 0.9, and 1.4%. The 0.9% *T. fuciformis* powder addition was selected as the most beneficial condition by sensory evaluation. The resulting ice cream was sweet but not greasy, with nice flavor and texture, and this nutritious, novel, and healthy product with a smooth taste was the most popular formula. It is exceptionally interesting that the control ice cream (i.e., milk ice cream) without any added *T. fuciformis* was also extraordinarily popular among all of the subjects. It is no wonder that milk ice cream is always one of the main commercial ice cream products, but it does not offer the functional nutrients of the *T. fuciformis* ice cream such as protein, polysaccharides, and dietary fiber.

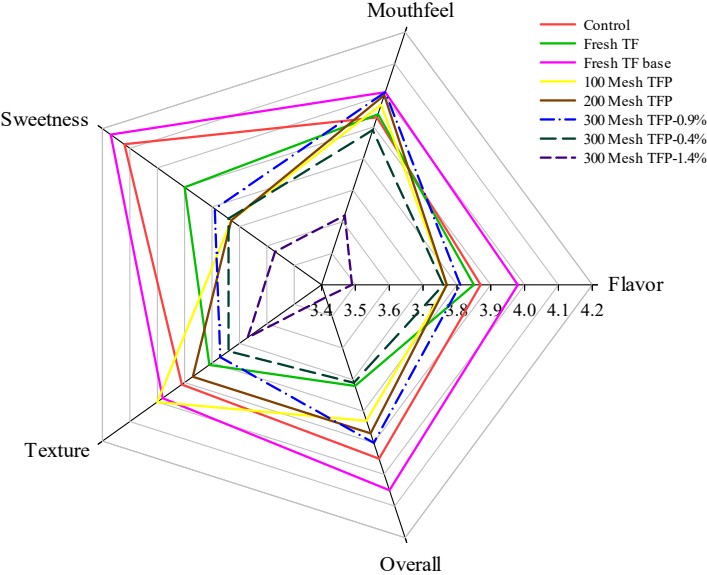

**Figure 2.** Sensory scores of *T. fuciformis* ice creams.

### 3.2. Melting Behavior and Melting Rate of T. fuciformis Ice Cream

Figure 3 exhibits 100 g of ice cream which was placed on an iron plate at 24 ± 0.5 °C for total testing time of 90 min. After 60 min, as long as the ice cream was supplemented with *T. fuciformis*, the ice cream was superior to the control ice cream. We observed that the fresh TF, fresh TF base, and

the 0.9% TFP enabled better integrity for all of the ice creams after 90 min in this study compared to the control sample. The particle size of the *T. fuciformis* powder and the proportions of the added amounts are not extraordinarily clear. From Figure 4, we obtained lower values of 0.37 and 0.39 g/min, respectively, for the fresh TF and fresh TF base among the ice creams. The slower the melting rate, the more resistant to melting the *T. fuciformis* ice cream became due to the fresh base; the fruiting body and the *T. fuciformis* powder ice cream exhibited anti-melting properties, but melting was extremely fast for the control ice cream. The resistance to melting is an important factor in the eating of ice cream. The addition of *T. fuciformis* had the effect of slowing the melting of the ice cream. Comparisons of Figures 3 and 4 are unable to subdivide the exact anti-melting effects of adding *T. fuciformis* powder to the ice cream. As shown in Figure 4, the fresh TF and fresh TF base were indeed the best; however, from Figure 3, the ice creams with fresh TF, fresh TF base and 300 mesh *T. fuciformis* powder added to 0.9% still maintained a certain shape after 90 min, whereas the other ice creams were almost liquefied. It is impossible to precisely ascertain the melting properties and melting rate features. This study must employ micro kinetics analysis for further detailed analysis.

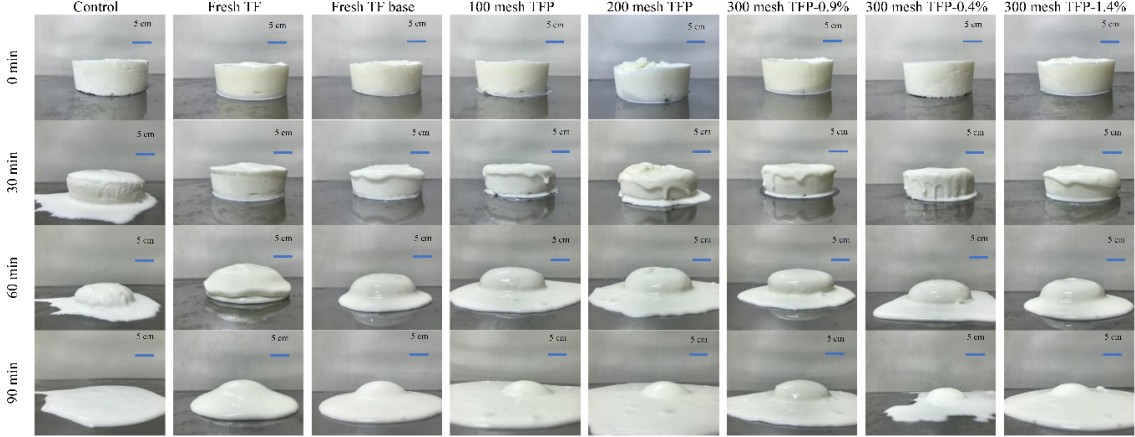

**Figure 3.** Photographs taken during melting behavior of ice creams produced with control, fresh TF, fresh TF base, 100 mesh TFP, 200 mesh TFP, 300 mesh TFP-0.9%, 300 mesh TFP-0.4% and 300 mesh TFP-1.4%. They were taken at the beginning of the test (0 min), and after 30, 60 and 90 min at room temperature.

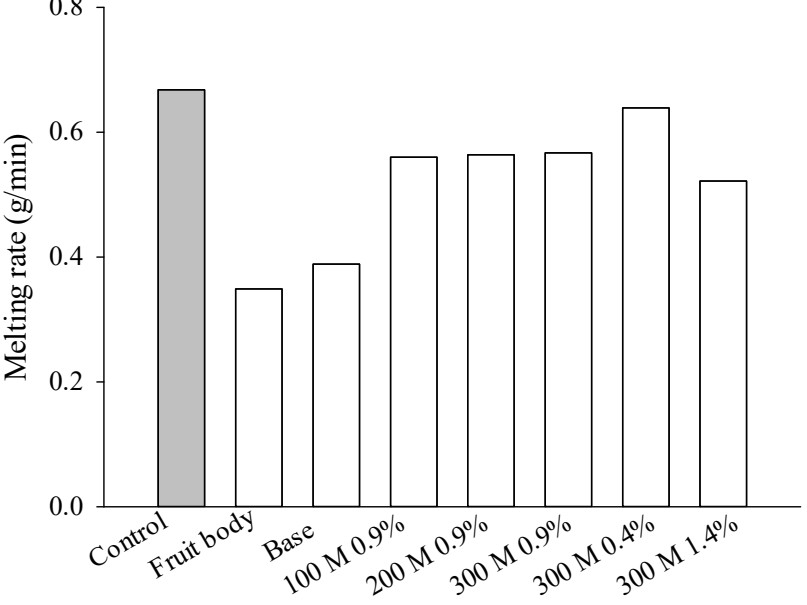

**Figure 4.** Melting rate of *T. fuciformis* ice creams.

### 3.3. Melting Properties of T. Fuciformis Ice Cream by DSC Analysis

First, in the study of *T. fuciformis* ice cream, we attempted to analyze the melting characteristics of commercial ice cream. Commercial ice cream was labeled with a composition ratio of 18% sucrose. From Table 2, the sucrose content ratios in this study ranged from 7.4 to 8.4%; due to the fresh base and the fruiting body of *T. fuciformis*, a water content of up to 90% existed, requiring an additional 10 g of sugar per process in response to the actual taste. Table 2 shows the control ice cream, which contained the lowest sucrose content of 7.5%, for which the glass transition temperature was unable to be detected by DSC analysis; however, this sample exhibited the highest melting onset temperature (Tg) of −2.88 °C, melting peak temperature ($mT_0$) of −0.37 °C, and enthalpy of the melting reaction ($m\Delta H$) of 196.20 kJ/kg. When sucrose content exceeded 10%, the glass transition temperature occurred at approximately −33 °C. The highest additional ratio, 50% sucrose, exhibited the lowest melting enthalpy of 75.65 kJ/kg, the glass transition temperature of −34.79 °C, melting onset temperature of −13.92 °C and a peak temperature of −6.95 °C in this study. DSC result shows that an increase in the proportion of sucrose added was detrimental to the stability of the frozen dessert. The higher the additional sucrose ratio of the cold dessert, the faster the ice melting. It is no wonder that gelato has added glucose or fructose to replace a fraction of the sucrose for increasing the rich taste, but that can also reduce the melting rate.

**Table 2.** Results of control ice cream, commercial ice cream, sugar solution of at 10, 30, and 50% sucrose by DSC analyses for heating from −60 to 30 °C with the heating rate 6/°C min.

| Sample | $T_g$ (°C) | $mT_o$ (°C) | $mT_p$ (°C) | $m\Delta H$ (kJ/kg) |
|---|---|---|---|---|
| 7.5% sucrose control ice cream | N/A | −2.88 | −0.37 | 196.20 |
| 18% sucrose commercial ice cream | −33.65 | −5.84 | −2.15 | 137.50 |
| 10% sucrose solution | −32.93 | −4.72 | −0.59 | 176.90 |
| 30% sucrose solution | −32.50 | −8.66 | −3.29 | 122.80 |
| 50% sucrose solution | −34.79 | −13.92 | −6.95 | 75.65 |

Legend: Tg, glass transition temperature; mTo, melting reaction onset temperature; mTp, Melting reaction peak temperature; m$\Delta$H, endothermic enthalpy of the melting; N/A, not applicable.

Regarding the determination of *T. fuciformis* ice cream melting properties by DSC analysis, which conducted dynamic tests of 2, 4, and 6 °C/min, the heating rates across the range of temperature increased after holding for 5 min at −60 °C and then increased to 30 °C for each melting experiment. The DSC analysis results could not determine whether the added proportion of Tremella fungus would affect the stability of ice cream (Table 3). It is not only a lack of functionality, but also the quantitative analysis results from the DSC curve that could be insufficient in providing the detailed differences between stabilizers added into the ice cream. Thus, Table 3 displays that *T. fuciformis* ice creams exhibit nearly identical results of melting onset temperature, melting peak temperature, and enthalpy of melting reaction by DSC preliminary analysis.

**Table 3.** Results of DSC tests of ice cream with heating rates of 2, 4, and 6 °C/min over the range of temperatures from −60 to 30 °C.

| | Heating Rate (°C/min) | $mT_o$ (°C) | $mT_p$ (°C) | $m\Delta H$ (kJ/kg) |
|---|---|---|---|---|
| Control | 2 | −2.91 | −0.96 | 202.50 |
| | 4 | −3.20 | −0.70 | 199.30 |
| | 6 | −2.88 | −0.37 | 196.20 |
| Fresh TF | 2 | −3.11 | −0.96 | 202.00 |
| | 4 | −3.09 | −0.47 | 209.10 |
| | 6 | −3.34 | −0.26 | 206.60 |
| Fresh TF base | 2 | −3.36 | −1.11 | 198.30 |
| | 4 | −3.26 | −0.63 | 201.70 |
| | 6 | −3.18 | −0.25 | 203.50 |

**Table 3.** *Cont.*

|  | Heating Rate (°C/min) | $mT_o$ (°C) | $mT_p$ (°C) | $m\Delta H$ (kJ/kg) |
|---|---|---|---|---|
| 100 mesh TFP | 2 | −3.03 | −0.90 | 203.10 |
|  | 4 | −3.25 | −0.70 | 203.90 |
|  | 6 | −3.30 | −0.43 | 203.40 |
| 200 mesh TFP | 2 | −2.89 | −0.86 | 197.60 |
|  | 4 | −3.09 | −0.62 | 201.40 |
|  | 6 | −3.16 | −0.32 | 201.40 |
| 300 mesh TFP-0.9% | 2 | −2.97 | −0.95 | 201.00 |
|  | 4 | −3.03 | −0.64 | 200.10 |
|  | 6 | −3.11 | −0.26 | 211.90 |
| 300 mesh TFP-0.4% | 2 | −2.72 | −0.79 | 209.00 |
|  | 4 | −2.85 | −0.45 | 212.90 |
|  | 6 | −2.79 | −0.19 | 215.30 |
| 300 mesh TFP-1.4% | 2 | −2.99 | −0.95 | 196.40 |
|  | 4 | −3.33 | −0.66 | 202.50 |
|  | 6 | −3.24 | −0.19 | 209.60 |

Legend: $mT_o$, melting reaction onset temperature; $mT_p$, Melting reaction peak temperature; $m\Delta H$, endothermic enthalpy of the melting; N/A, not applicable.

### 3.4. Ice Cream Melting Kinetic Simulation

The melting reaction of *T. fuciformis* ice cream has not been addressed by much applicable literature to inform or assist this study. According to the authors' previous research experience, the nonchemical reaction kinetic was selected, which was performed by Avrami Erofeev's kinetic and the proto-kinetic equations simulation analyses of the phase transfer, melting, and endothermic reaction, of the physical properties. Here, we used DSC analysis with various heating rates to compute the melting kinetics of ice cream by conducting Avrami Erofeev's kinetic and proto-kinetic simulations, to glean all kinetic model simulation results and then to select and compare the more suitable kinetic model for the application of ice cream melting characteristics. Furthermore, we finished all of the ice cream melting reaction kinetic calculations, which included the melting property parameters, such as the natural logarithm of the pre-exponential factor, the apparent activation energy of the melting reaction, and the reaction order. The detailed *T. fuciformis* ice cream melting reaction parameters are in Table 4.

Table 4 shows the Avrami Erofeev kinetic equation simulation, which exhibits poorly consistent results. In addition, Figure 5 also displays the results of the proto-kinetic model simulation; all of the simulation results coincide with the experimental data, but using Avrami Erofeev's kinetic model results in poor overall consistency. Furthermore, the results of the proto-kinetic model simulation were applied to compare the differences of melting characteristics of *T. fuciformis* ice creams in this study. From Table 4, we obtained the fresh TF and fresh TF base *T. fuciformis* ice creams with almost identical kinetic parameters. For the result of adding *T. fuciformis* powder as an ice cream stabilizer, compared with the melting parameters of various particle sizes (100, 200, and 300 mesh), the 300 mesh displayed a more stabilized effect. From the results of the kinetic analysis, the controlled ice cream exhibited the worst resistance to melting.

**Table 4.** Melting kinetic simulation results of DSC analyses for *T. fuciformis* ice creams from −60 to 30 °C at heating rates of 2, 4, and 6 °C/min.

| Ice Cream | Heating Rate | 2 | | 4 | | 6 | |
|---|---|---|---|---|---|---|---|
| | Parameter | Avrami Erofeev's | Proto | Avrami Erofeev's | Proto | Avrami Erofeev's | Proto |
| Control | $\ln(k_0)$ | 1.000E-06 | 3.0478 | 30.4182 | 3.3370 | 12.2110 | 3.6327 |
| | $E_a$ | 18.2713 | 16.3207 | 76.0868 | 15.8026 | 36.0000 | 15.8015 |
| | $n/n_1$ | 0.3944 | 1.0082 | 0.8008 | 0.9998 | 0.7618 | 0.9987 |
| | $n_2$ | N/A | 0.0909 | N/A | 0.3000 | N/A | 0.3278 |
| | $\Delta H$ | 205.8837 | 204.2369 | 200.0942 | 201.9552 | 200.2802 | 200.4360 |
| Fresh TF | $\ln(k_0)$ | 43.5299 | 23.3498 | 19.8606 | 24.3940 | 47.0421 | 24.1891 |
| | $E_a$ | 106.2144 | 62.6209 | 54.2131 | 62.4601 | 116.3713 | 61.3940 |
| | $n/n_1$ | 0.8283 | 0.7573 | 0.6491 | 0.8687 | 0.1507 | 0.8505 |
| | $n_2$ | N/A | 0.2359 | N/A | 0.4905 | N/A | 0.6690 |
| | $\Delta H$ | 200.7589 | 201.4170 | 209.7087 | 209.0389 | 204.5348 | 203.0871 |
| Fresh TF base | $\ln(k_0)$ | 8.3507 | 23.9527 | 23.8063 | 24.4070 | 47.4876 | 23.9588 |
| | $E_a$ | 27.0063 | 62.5884 | 61.0614 | 62.7213 | 114.0540 | 62.3057 |
| | $n/n_1$ | 0.9569 | 0.8875 | 0.8451 | 0.8391 | 0.7214 | 0.6975 |
| | $n_2$ | N/A | 0.3678 | N/A | 0.5185 | N/A | 0.4523 |
| | $\Delta H$ | 194.3651 | 196.5890 | 199.2232 | 199.6637 | 201.3932 | 204.5091 |
| 100 mesh TFP | $\ln(k_0)$ | 10.2859 | 10.4017 | 7.9790 | 10.6846 | 32.4467 | 11.0755 |
| | $E_a$ | 31.3031 | 31.9875 | 28.2942 | 31.9423 | 79.8970 | 31.3979 |
| | $n/n_1$ | 0.9525 | 0.9872 | 0.7214 | 0.9323 | 0.7972 | 0.9828 |
| | $n_2$ | N/A | 0.4203 | N/A | 0.4366 | N/A | 0.5828 |
| | $\Delta H$ | 204.9092 | 203.7050 | 204.6270 | 204.6387 | 203.7039 | 203.3011 |
| 200 mesh TFP | $\ln(k_0)$ | 28.3391 | 19.6084 | 36.7518 | 19.2370 | 11.6076 | 19.8417 |
| | $E_a$ | 71.8479 | 52.5000 | 90.7832 | 52.3321 | 35.2615 | 51.1932 |
| | $n/n_1$ | 0.8875 | 0.9407 | 0.7486 | 0.7454 | 0.6929 | 0.9170 |
| | $n_2$ | N/A | 0.3689 | N/A | 0.1915 | N/A | 0.6586 |
| | $\Delta H$ | 195.3120 | 195.6301 | 201.8613 | 203.1085 | 204.6257 | 203.7534 |
| 300 mesh TFP-0.9% | $\ln(k_0)$ | 40.9786 | 25.0972 | 14.8281 | 25.0105 | 15.8715 | 27.8490 |
| | $E_a$ | 100.4681 | 65.4591 | 42.6276 | 64.1818 | 43.3537 | 69.4753 |
| | $n/n_1$ | 0.8353 | 0.8577 | 0.7291 | 0.8086 | 0.8278 | 0.8497 |
| | $n_2$ | N/A | 0.3644 | N/A | 0.8310 | N/A | 0.6155 |
| | $\Delta H$ | 200.5455 | 200.7974 | 199.3297 | 200.1815 | 212.2757 | 210.7239 |
| 300 mesh TFP-0.4% | $\ln(k_0)$ | 29.5765 | 14.7524 | 17.2494 | 14.8923 | 29.9857 | 14.3501 |
| | $E_a$ | 74.6918 | 41.1389 | 48.1955 | 41.2655 | 74.5872 | 38.7238 |
| | $n/n_1$ | 0.8821 | 1.0046 | 0.6937 | 0.9120 | 0.8006 | 0.9589 |
| | $n_2$ | N/A | 0.4052 | N/A | 0.6784 | N/A | 0.6391 |
| | $\Delta H$ | 208.6359 | 208.4884 | 216.9507 | 211.5991 | 207.9163 | 214.1946 |
| 300 mesh TFP-1.4% | $\ln(k_0)$ | 12.2034 | 25.2985 | 11.8881 | 26.1150 | 29.1130 | 26.3340 |
| | $E_a$ | 38.4980 | 66.4692 | 36.1361 | 66.3921 | 72.7000 | 66.3508 |
| | $n/n_1$ | 0.7198 | 0.7998 | 0.7568 | 0.8430 | 0.7894 | 0.8206 |
| | $n_2$ | N/A | 0.2044 | N/A | 0.5349 | N/A | 0.6323 |
| | $\Delta H$ | 197.3906 | 196.7799 | 200.5685 | 204.5810 | 208.5698 | 208.6545 |

$\ln(k_0)$, natural logarithm for the pre-exponential factor of reaction($\ln 1/s$); $E_a$, activation energy of the melting reaction (kJ/mol); $n/n_1$, reaction order $n_1$ of the melting reaction; $n_2$, reaction order $n_2$ of the melting reaction. Enthalpy of the melting reaction by simulation (kJ/kg); $\Delta H$, enthalpy of the melting reaction by simulation (kJ/kg); Avrami Erofeev's, Avrami Erofeev's kinetic equation; Proto, Proto-kinetic equation.

The above results match well with the actual melting rate measurement results. In addition, comparing the various exceptional addition ratios of 300 mesh *T. fuciformis* powder (0.4, 0.9, and 1.4%), it is shown that the higher the amount of 1.4% addition, the better the stability. Even when the addition amount is high, the anti-melting property is good, but this study had to consider the results of the sensory evaluation. Moreover, from Table 4, we selected the results of the proto-kinetic model simulation comparison, which indicated the fresh TF, fresh TF base and the 0.9% *T. fuciformis* powder; there were exceptionally similar melting kinetic parameters. From the comparison of sensory

evaluation and melting kinetic analysis results, it can be demonstrated that the best addition ratio of *T. fuciformis* powder was 0.9% and that the overall properties were the most suitable with the fresh base and the fruiting body ice cream in this study.

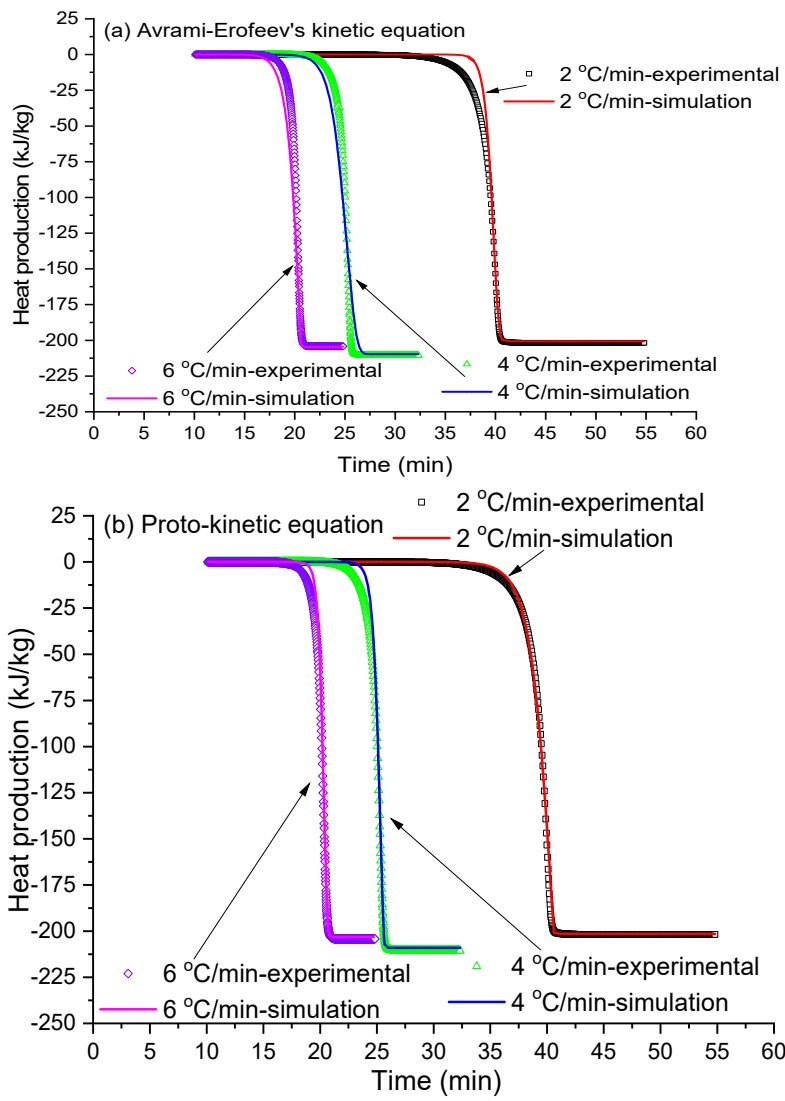

**Figure 5.** Selected fresh TF ice cream heat production versus time curves with DSC various heating rates of 2, 4, and 6 °C/min by comparing (**a**) Avrami-Erofeev's kinetic and (**b**) Proto-kinetic model simulation.

### 3.5. Physicochemical Qualities of T. Fuciformis Ice Cream

Physical characteristics and chemical composition of *T. fuciformis* ice cream are listed in Table 5. There are no significant differences in pH, L value, a value, WI value, ash, fat and water-insoluble dietary fiber between the two fresh TF base and 300 mesh TFP-0.9% ice creams. In all analysis results, the WI value is an important quality characteristic of products that influences the food choices of consumers. It was very sure, that mixing in various *T. fuciformis* powder did not affect the appearance of ice cream in this study. Moreover, adding *T. fuciformis* will increase the viscosity and decrease the hardness of ice cream. Here, *T. fuciformis* powder and water have an affinity with each other to inhibit the formation of large ice crystals which affects the texture state and also enhances the melting resistance of ice cream [1]. *T. fuciformis* is rich in polysaccharides; the viscosity will increase when added to ice cream [26]. Because *T. fuciformis* is rich in polysaccharides, when it is added into ice cream, the viscosity of ice cream can be increased. However, the protein, water-soluble dietary fiber, and polysaccharide of the fresh TF base and 300 mesh TFP ice cream are higher content than the control ice

cream due to the addition *T. fuciformis*. The components such as *T. fuciformis* were added to the ice cream to increase the functionality and improve the nutritional value.

**Table 5.** Physical characteristics and chemical composition of *T. fuciformis* ice creams.

|  | Control | Fresh TF Base | 300 Mesh TFP-0.9% |
|---|---|---|---|
| Physical characteristics |  |  |  |
| pH | 6.91 ± 0.03a | 6.69 ± 0.05b | 6.80 ± 0.12ab |
| Color coordinate L | 82.36 ± 2.77a | 81.98 ± 0.76a | 79.54 ± 0.42a |
| a | 3.01 ± 0.29a | 3.38 ± 0.18a | 3.37 ± 0.02a |
| b | 8.66 ± 0.48b | 10.89 ± 1.08a | 10.38 ± 0.91ab |
| WI | 82.84 ± 1.27a | 82.77 ± 2.74a | 80.29 ± 0.88a |
| Hardness (N/mm$^2$) | 3.44 ± 0.53a | 1.67 ± 0.49b | 2.33 ± 0.48b |
| Viscosity (cp) | 35.40 ± 0.11c | 558.05 ± 10.25a | 410.10 ± 17.98b |
| Chemical composition |  |  |  |
| Dry matter (%) | 26.08 ± 1.01c | 43.88 ± 1.87a | 39.95 ± 1.68b |
| Ash (DM, %) | 0.51 ± 0.01a | 0.50 ± 0.01a | 0.50 ± 0.02a |
| Fat (DM, %) | 55.02 ± 3.31a | 53.67 ± 2.66a | 54.99 ± 3.03a |
| Protein (DM, %) | 16.13 ± 0.38c | 17.84 ± 0.15a | 17.24 ± 0.25b |
| Water-soluble dietary fiber (DM, %) | 0.09 ± 0.02b | 0.19 ± 0.02a | 0.16 ± 0.01a |
| Water-insoluble dietary fiber (DM, %) | 0.23 ± 0.06a | 0.27 ± 0.04a | 0.24 ± 0.04a |
| Polysaccharide (DM, mg/g) | 113.40 ± 2.38c | 200.50 ± 1.31b | 281.01 ± 5.48a |

L, lightness, a, greenness/redness, b, blueness/yellowness, WI, whiteness index; DM, dry matter. 1 replicate of each of 3 independently treated samples and each value is expressed as mean ± SD. Means with different letters within a column differ significantly ($P < 0.05$).

## 4. Conclusions

A sensory evaluation was performed among ice creams with fresh fruit body, fresh base, and 300 mesh powder of *T. fuciformis* as a stabilizer, to determine which generated a better taste. The addition of *T. fuciformis* powders of various particle sizes was compared to the sensory evaluation of ice cream and anti-melting characteristics, and the addition of 300 mesh powder yielded a better taste. Furthermore, we compared the amount of 300 mesh powder suitable for the addition and the addition ratio at 0.9% produced a better mouthfeel. From the results of the melting kinetic analysis, it was again proven that the fresh fruit body, the fresh base, and the 0.9% additive ratio of 300 mesh powder of *T. fuciformis* ice cream presented exceptionally similar melting kinetic parameters. Meanwhile, the higher the amount of sucrose added, the worse the anti-melting stability of the ice cream; when the addition was 10% or more, the ice product could be measured ca. −33 °C, and the glass transition phenomenon appeared. This can be applied to the processing of ice dessert with respect to the addition of sucrose and the anti-melting relation. Overall, we used Avrami–Erofeev's kinetic and Proto-kinetic model simulations to successfully determine the differences in the kinetic parameters of ice cream melting with the addition of various *T. fuciformis* stabilizers. This information can be provided to food processing industries for the improvement and development of frozen dessert processing technology.

**Author Contributions:** Data curation, G.J.T. and C.-Y.L.; Conceptualization, S.-Y.T.; Methodology, C.-P.L.; Y.-T.H.; Writing—original draft preparation, C.-P.L.; Writing—review and editing, S.-Y.T. All authors have read and agreed to the published version of the manuscript.

**Funding:** The authors are grateful to the Ministry of Science and Technology (MOST) and Asia University (ASIA) under the contract No.: MOST 106-2221-E-468-013-, 107-2221-E-468-007-MY2 and ASIA-108-CMUH-02.

**Acknowledgments:** We are grateful to Mid-Sum Biotechnology Co. Ltd. for providing the *Tremella fuciformis*.

**Conflicts of Interest:** The authors declare no conflict of interest.

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
