# Peer review of "Assessment of Melting Kinetics of Sugar-Reduced Silver Ear Mushroom Ice Cream under Various Additive Models"

_applsci, doi:10.3390/app10082664_

Round 1

Reviewer 1 Report

Νο ψομμεντ

Author Response

Thank you for reviewing our paper.

Reviewer 2 Report

Manuscript "Assessment of Melting Kinetics of Sugar-reduced Silver Ear Mushroom Ice Cream under Various Additive Models" is well written. Introducion is compact and comprehensive. The methodology and results are described in a clear and complete way. Only one thing is unclear. Why did the Authors test only 300 mesh powder in various ratios?

Author Response

Please see p. 2, Line 76-78: the previous studies by the authors, which exhibited  the complete analysis results of the thermal stability of the particles and the greater the moisturizing capability compared to the larger particles (300 mesh) via DSC tests.

Reviewer 3 Report

The paper entitled „Assessment of Melting Kinetics of Sugar-reduced Silver Ear Mushroom Ice Cream under Various Additive Models” focuses on assessing the effects of various food processing silver ear (Tremella fucitormis) powders in sugar-reduced ice cream through melting kinetic simulation, sensory properties and functional component.seems interesting and can be useful for the potential consumers. This is a trial to prove the effectivensess of novel melting method together with addition of T. fuciformis into the consistency of ice-cream. The subject can be slightly interesting for a potential reader, but authors should improve the paper to fulfill high standard of Applied Sciences Journal.

Minor comments:

  1. There is no reference to Table 1.
  2. Please check the numbers in the formulations (Tab. 1). It seems that Silver eat addition is 4,83, not 4,85 (0,9%), 2,13 instead of 2,15 (0,4%). Please change the order in receipe in the ascending order (0,4; 0,9; 1,4%)
  3. 173 – ‘ in ice cream’
  4. 262 – ‘from Table 3, they cannot determine…’ – rewrite
  5. In table 5 results for all analyzed types of ice-creams should be included

Major comments:

  1. The statement from the abstract (l.30) ‘Overall, the results and experience of this research can be provided as a reference for the frozen dessert processing technology’ should be justified in the text and results.
  2. The introduction should be extended by the overview of currently known developed stabilizers as stated in l. 47.
  3. In l. 62 authors mentioned the new methodology developed by them. The description of the method is needed followed by references or at least by the comparison to other known metohods to justify the use of that method during the research analyzed in the paper.
  4. Statistical significance is missing for Fig. 2, Fig. 4, Fig. 5, Tab. 2, Tab. 3, Tab. 4, Tab 5.
  5. Without statistical significance results it is impossible to follow discussion and conlusions.

Author Response

Comments from reviewer 3

Response

3) The paper entitled “Assessment of Melting Kinetics of Sugar-reduced Silver Ear Mushroom Ice Cream under Various Additive Models” focuses on assessing the effects of various food processing silver ear (Tremella fuciformis) powders in sugar-reduced ice cream through melting kinetic simulation, sensory properties and functional component. seems interesting and can be useful for the potential consumers. This is a trial to prove the effectiveness of novel melting method together with addition of T. fuciformis into the consistency of ice-cream. The subject can be slightly interesting for a potential reader, but authors should improve the paper to fulfill high standard of Applied Sciences Journal.

Thank you for reviewing our paper. We have made appropriate corrections for meeting your expectations, as you can see from the revised version.

Minor comments:

1.     There is no reference to Table 1.

2.     Please check the numbers in the formulations (Tab. 1). It seems that Silver eat addition is 4,83, not 4,85 (0,9%), 2,13 instead of 2,15 (0,4%). Please change the order in recipe in the ascending order (0,4; 0,9; 1,4%)

3.     173 – ‘ in ice cream’

4.     262 – ‘from Table 3, they cannot determine…’ – rewrite

5.     In table 5 results for all analyzed types of ice-creams should be included

1. Table 1 is the formula table of ice cream, which is selected for according to the preliminary test of ice cream. Thus, there is no reference.

2. We have rewritten the description, please see Table 1 in the current text.

3. We have rewritten the description, please see lines173 in the current text.

4. We have rewritten the description in the current text. Please see lines 262-263.

5. Due to the large number of samples, the selection of the most representative samples for physical and chemical analysis is sufficient to represent the results of this study.

Major comments:

1.         The statement from the abstract (L. 30) ‘Overall, the results and experience of this research can be provided as a reference for the frozen dessert processing technology’ should be justified in the text and results.

2.         The introduction should be extended by the overview of currently known developed stabilizers as stated in L. 47.

3.         In L. 62 authors mentioned the new methodology developed by them. The description of the method is needed followed by references or at least by the comparison to other known methods to justify the use of that method during the research analyzed in the paper.

4.         Statistical significance is missing for Fig. 2, Fig. 4, Fig. 5, Tab. 2, Tab. 3, Tab. 4, Tab 5.

5.         Without statistical significance results it is impossible to follow discussion and conclusions.

1. We have rewritten the description in the current text. Please see lines 30-32.

2. We have rewritten the description in the current text. Please see lines 47.

3. According your comments, we have added the references and the description in the current text. Please see lines 64.

4. Figure 2 is a radar chart expressing sensory results, which is already the result of statistical analysis. Figure 4 is also a figure expressed by the results of statistical analysis.

5. The following is a reply to your questions:

1)     This study was processed with the kinetics evaluated by applying thermal safety software (including ARKS TA and ARKS FK software) developed by ChemInform Saint-Petersburg (CISP) Ltd, http://www.cisp.spb.ru. The method for the creation of a kinetic model and the algorithms that are employed are clearly described on the homepage.

2)     One general rule of principal importance–the kinetic model must be evaluated from the whole set of available experimental data. If there are several tests at different temperatures or different concentrations, then parameters should be estimated when all the data are included together. Therefore, the best parameters are those that provide better fitting of all the tests. Apparently, each separate experiment can be fitted better and parameters for different tests (temperature or concentrations will be different), but the resultant kinetic model must be based on all the experiments.

Thank you for your suggestion, we will be basing this on your recommendation in the future.

Round 2

Reviewer 3 Report

I accept this manuscript in present form